SMARTAR: an R package for designing and analyzing Sequential Multiple Assignment Randomized Trials

Zhong Xiaobo xiaobo.zhong@mountsinai.org 1
Cheng Bin 2
Wang Xinru 2
Cheung Ying Kuen 2
1 Department of Population Health Science and Policy, Icahn School of Medicine at Mount Sinai , New York , NY , USA
2 Department of Biostatistics, Columbia University , New York , NY , USA
Aly Sharif
Electronic publication date: 2021 Jan 11
Publication date: 2021
Volume: 9
Electronic Location ID: e10559
Received 2020 Apr 21; Accepted 2020 Nov 22
Copyright: ©2021 Zhong et al.
Copyright year: 2021
Copyright holder: Zhong et al.
License: This is an open access article distributed under the terms of the Creative Commons Attribution License, which permits unrestricted use, distribution, reproduction and adaptation in any medium and for any purpose provided that it is properly attributed. For attribution, the original author(s), title, publication source (PeerJ) and either DOI or URL of the article must be cited.
License URL: https://creativecommons.org/licenses/by/4.0/

Keywords: Adaptive Treatment Strategy, Analysis, Clinical Trial, Design, R package, SMART

Funding: National Cancer Institute P30 CA196521 This work was supported by the National Cancer Institute (P30 CA196521). The funders had no role in study design, data collection and analysis, decision to publish, or preparation of the manuscript.

==============================
This article introduces an R package, SMARTAR (Sequential Multiple Assignment Randomized Trial with Adaptive Randomization), by which clinical investigators can design and analyze a sequential multiple assignment randomized trial (SMART) for comparing adaptive treatment strategies. Adaptive treatment strategies are commonly used in clinical practice to personalize healthcare in chronic disorder management. SMART is an efficient clinical design for selecting the best adaptive treatment strategy from a family of candidates. Although some R packages can help in adaptive treatment strategies research, they mainly focus on secondary data analysis for observational studies, instead of clinical trials. SMARTAR is the first R package provides functions that can support clinical investigators and data analysts at every step of the statistical work pipeline in clinical trial practice. In this article, we demonstrate how to use this package, using a real data example.

Introduction

SMARTAR (Sequential Multiple Assignment Randomized Trial with Adaptive Randomization) is an R package that provides functions for primary data analysis and the design calibration of sequential multiple assignment randomization trials (SMARTs) to compare multiple adaptive treatment strategies (ATSs). An ATS is a multistage treatment procedure consisting of a sequence of decision-making rules, one per treatment decision, that allows for the selection of a treatment, based on the clinical information collected by each decision-making time point (Kosorok & Moodie, 2015). An ATS is alternatively called a dynamic treatment regime (Murphy et al., 2001), a treatment policy (Lanceford, Davidian & Tsiatis, 2002), or an adaptive intervention (Collins, Murphy & Bierman, 2004). In comparison to traditional intervention, under which a patient receives a fixed treatment throughout the entire course, in ATS the treatment level and type can be repeatedly adjusted according to the individual’s needs, providing opportunities to optimize the treatment selection as a function of time-dependent clinical information. Therefore, the concept of ATS fits the paradigm of personalized medicine. ATS is particularly effective for managing chronic conditions, such as substance disorder, depression management, and cancer.

Figure. 1 gives Murphy’s (2005) example of a two-stage ATS for controlling heavy drinking addiction. Under this strategy, alcohol-dependent patients were first given intensive cognitive behavioral therapy (CBT) for two months in stage 1. At the end of stage 1, the patients were classified as either responders or non-responders, according to their level of heavy drinking as recorded by clinicians. The decision for stage-2 treatment was adapted to the intermediate response. Non-responders were switched to the medication of opiate antagonists (MED). On the other hand, responders were recommended telephone monitoring and counseling (TMC). Comparing to a simple nonadaptive strategy, under which all the treatment decisions are made at baseline, this ATS provides an opportunity to adjust the stage-2 treatment based on the stage-1 treatment and intermediate response of the same patient. Thus, ATS can potentially improve the treatment benefit in the long run.

Figure 1 Two-stage adaptive treatment strategy for addiction management.

SMART is a randomized clinical trial (RCT) design aimed at selecting the optimal ATS, defined as the strategy leading to the best average value of the primary outcome. A distinct feature of SMART is that it assigns patients to different treatment arms across multiple stages by sequential randomization. Figure 2A shows a SMART for alcohol addiction management in which the patients were first randomized between two treatment options (i.e., CBT and MED) at baseline and then re-randomized between four treatment options at stage 2, according to the history of stage-1 treatments and intermediate responses. For example, a patient who responds to CBT is randomized to either TMC or TM, whereas a non-responder to MED is randomized to MED or a combination of the enhanced motivational program (EM), CBT, and MED. As a result, the trial provides data on 8 ATSs as Table 1, including the previous ATS example in Fig. 1.

Figure 2 (A) A SMART design for comparing two-stage ATSs for addiction (R, randomization; CBT, cognitive behavioral therapy; MED, medication; TMC, telephone monitoring and counseling; TM, telephone monitoring). (B) Example of a degenerate SMART design.

Table 1 Two-stage adaptive treatment strategies defined in an alcohol addiction trial.

		Stage-1	Stage-2 treatment	
Strategy	Code	treatment	Respondent	Non-respondent	
1	(1;3,2)	CBT	TMC	MED	
2	(1;3,5)	CBT	TMC	EM+CBT+MED	
3	(1;4,2)	CBT	TM	MED	
4	(1;4,5)	CBT	TM	EM+CBT+MED	
5	(2;3,1)	MED	TMC	MED	
6	(2;3,5)	MED	TMC	EM+CBT+MED	
7	(2;4,1)	MED	TM	MED	
8	(2;5,5)	MED	TM	EM+CBT+MED	
Notes.

CBT cognitive behavioral therapy

EM enhanced motivational program

MED medication

TM telephone monitoring

TMC telephone counseling and monitoring

SMART is an advanced early phase RCT design for investigators who need to select a promising intervention from multiple candidates and move it forward to a confirmatory trial. Compared to a classic design that only randomizes patients at baseline to a series of pre-specified strategies, SMART takes advantage of sequential randomization and is thus more efficient (Ko & Wahed, 2012). Suppose a study aims to evaluate the eight strategies in Table 1. A traditional RCT design randomizes patients at baseline to eight independent groups, each of which corresponds to an ATS of interest. To obtain n patients for each ATS, assuming equal randomization probabilities, we need 8n patients in total. However, under the SMART design in Fig. 2A, we can obtain data on 2n patients to evaluate each ATS, for a total sample size of 8n. SMART is being increasingly used for managing chronic diseases in which patients typically face many treatment profiles across multiple stages (e.g., cancer, depression, and substance disorder).

On the other hand, the sequential feature introduces challenges in analyzing SMART data. First, unlike most classic RCT designs, under which patients are randomized to be independent treatment groups, a patient who follows a specific treatment sequence in SMART can contribute information in the evaluation of multiple strategies. For example, a patient who received CBT at stage 1, failed to respond, and was switched to MED at stage 2 in Fig. 2 contributed data to evaluate both strategies 1 and 3 in Table 2. Such an overlap structure introduces covariance in statistical inference for comparison. Given SMART’s flexible design structure, most classic statistical methods used to analyze RCT cannot be directly applied to this setting (Murphy, 2005; Dawson & Lavori, 2012). Another potential issue is related to the dimensionality of the SMART data. The total number of ATSs embedded in a SMART increases exponentially as the design structure becomes more complex, which typically reflects more treatment stages, more treatment options, or more intermediate response categories. For example, if we add a treatment option to the stage-2 randomization for both responders and non-responders in Fig. 2, the total number of strategies embedded in this trial will increase from eight to 18. Therefore, it is crucial in SMART to control for false positive findings at a certain level (Ogbagaber, Karp & Wahed, 2016; Zhong et al., 2019). For more details on ATS and SMART, see Chakraborty & Murphy (2014).

Table 2 Primary functions built into the SMARTAR package.

Function	Outputs	
smartsize	Results of the sample size calculation	
getncp	Value of the noncentrality parameter of the chi-squared test statistics under the alternative hypothesis of the global test	
seqmeans	Design diagram, descriptive statistics, and summarized graphics at the sequence level	
atsmeans	Descriptive statistics and summarized graphs at the ATS level	
smartest	Results of the global and pairwise tests; output simultaneous confidence intervals for ATS comparisons	

The development of the SMARTAR pracakge was motivated by the need for a user-friendly comprehensive R package to help data analysts and clinical investigators complete the statistical work throughout the pipeline of the entire SMART, from design to data analysis, and then report generation. Although some R packages, including iqLearn (Linn, Laber & Stefanski, 2015), DTR (Tang & Melguizo, 2015), DTRreg (Wallace, Moodie & Stephens, 2017), and DynTxRegime (Holloway et al., 2018), were created for ATS-related research, they all focus on secondary analysis for observation studies, instead of SMARTs. These methods are not commonly used in clinical trial analysis and do not support sample size calculations, which is crucial in designing a clinical trial. By the time SMARTAR was developed, the only available R package on the Comprehensive R Archive Network (CRAN; see R Core Team, 2020) that supported sample size calculations of SMARTs was smartsizer (Artman, 2019). However, this package has two limitations. First, it provides results only generated by Monte Carlo simulations but not based on any available statistical inferential method. Second, it only supports sample size calculations based on a pairwise comparison approach. Thus, it cannot be applied to calculations based on a global comparison approach, which is important if the investigator lacks sufficient information about which pair of ATSs should be targeted at the design stage.

The SMARTAR package appeals to clinical investigators and data analysts in trial practice. Compared to ATS-related R packages currently available from CRAN, this package has the following innovative features. First, it supports exploratory data analysis (EDA), an efficient approach to summarizing the characteristics of data with visual methods and widely used in clinical trials. SMARTAR provides graphical tools at two different levels: the treatment sequence level and the strategy level. It can also help generate a SMART design diagram, which is essential for design purposes. Second, it is the first R package that can construct and directly output simultaneous confidence intervals for ATS comparisons. Simultaneous confidence intervals for group comparisons are commonly reported in clinical trials, wherein multiplicity adjustment is necessary. This package allows simultaneous confidence intervals to be output, corrected by Bonferroni’s method (Dunn, 1961). Third, SMARTAR provides the results of sample size calculations based on various published statistical methods in two different ways, with a global test and with a pairwise test. An investigator can then compare the results based on various statistical methods, including a global test based on inverse probability weights (IPWs; see Ogbagaber, Karp & Wahed, 2016), a likelihood-based global test (Zhong et al., 2019), an IPW-based pairwise test (Murphy, 2005), and a likelihood-based pairwise test (Dawson & Lavori, 2010). Such comparisons help maximize the design efficiency in trial practice.

In the remainder of this article, we first introduce some fundamental concepts and notation in designing and analyzing SMARTs, and then briefly introduce the statistical methods built in this package. We then present a general introduction of five primary functions in this package, followed by how to implement them, using a real data example.

Materials and Methods

Important concepts and notation

We introduce some fundamental concepts and notation of ATS under the SMART framework to help further illustration. We define a stage as an interval of time during which a patient receives only one treatment. At the beginning of stage t, where t = 1, …, T and T is the total number of stages in a SMART, an evaluation Oti is made to the ith patient. Here, i = 1, …, n is the patient index and n is the total sample size. Let Ati be the treatment given to patient i at stage t and let Y i be the final primary outcome observed at the end of the study. Data collected from subject i who completes a T-stage SMART can be summarized as a longitudinal trajectory: (1) O1i,A1i,…,OTi,ATi,Yi

For example, in the alcohol addiction trial example (Fig. 2A), considering that stage -1 treatment is not adaptive to baseline information, data collected from patient i can be summarized as A1i,O2i,A2i,Yi. In this case, A1i ∈ {CBT, MED} represents the stage-1 treatment a patient can possibly receive, A2i ∈ {TMC, TM, MED, EM + CBT + MED} represents the stage-2 treatment, O2i is the binary intermediate outcome measured at the end of stage-1, and Y i is the primary outcome defined as the percentage of days of abstinence over the entire treatment period. We use an overbar above the letter to indicate the history of that measurement up to the current time, when the event denoted by the letter occurs. For example, A ¯2i=A1i,A2i indicates the history of treatments received by patient i at stages 1 and 2 and O ¯2i=O1i,O2i represents the evaluation results of patient i up to O2i. We can thus simplify Eq. (1) as (2) O ¯Ti,A ¯Ti,Yi.

We suppress the subject indicator i for convenience whenever appropriate in the remainder of this article, so that Eqs. (1) and (2) become O1,A1,…,OT,AT,Y and O ¯T,A ¯T,Y, respectively. Let dtO ¯t,A ¯t−1 be the decision-making rules of the action of A ¯t conditioning on the history of O ¯t,A ¯t−1. An ATS can be viewed as a collection of decision-making rules across different clinical historical paths and can thus be defined as (3) D=d1O ¯1;⋯;dTO ¯T,A ¯T−1

wherein dtO ¯t,A ¯t−1, a scalar or a vector, is a stage-specific decision-making rule for selecting At and depends on the combination of O ¯t,A ¯t−1. In the alcohol addiction trial example, let {1,2,3,4,5} be the treatment options CBT, MED, TMC, TM, and EM+CBT+MED, respectively, and let {0,1} be the intermediate outcomes of response and no response. We can code the eight alcohol management strategies as shown in the second column of Table 1.

In practice, it is common but not necessary to define Ot as binary, to reduce the dimensionality of the SMART data. To simplify the design structure to improve efficiency, SMART also allows for the degenerate structure to reflect the knowledge of investigators from previous studies. For example, suppose an investigator already knows two factors: first, no treatment at stage 2 provides an additional effect for the initial treatment of MED; second, TMC is universally better than TM for responders to CBT. It is reasonable to degenerate the SMART design from Figs. 2A to 2B. For more details of degenerated SMART designs, see Zhong et al. (2019). SMARTAR can handle SMART with any design structure, including degenerate cases.

Statistical methods

We briefly introduce the methods commonly applied to SMART and included in this R package. Although trial practice always starts with a sample size calculation at the design stage, followed by data analysis. Given that the sample size calculation is always based on the method of the primary analysis, we first introduce the methods of comparison and then the sample size calculation. Given the aim of this article, we skip some technical details for simplicity. For more details, see the original articles of these methods.

Let θ be the value of an ATS, defined as the average primary outcome of patients across all the treatment sequences under the strategy. To estimate the ATS value, IPW estimatior is common, by which the weight of estimation is constructed based on the production of randomization probabilities corresponding to an individual (Murphy, 2005). Alternatively, we can use G-computation estimation (Lavori & Dawson, 2007), based on analyzing the marginal mean of the primary outcome following each treatment sequence under the ATS of interest.

Let θ ˆg be the estimated value of the gth ATS, where g is the strategy indicator and g = 1, …,G. For example, in the alcohol addiction example, the total number of strategies G = 8 (see Table 1). We can build a Wald-type global test with hypotheses: (4) H0:θ1=⋯=θGversusH1:θ′gsare not all equal for g=1,…G.

Let C = (1G−1| − IG−1) be a (G − 1) × G contrast matrix and let Σ ˆ be the consistent estimator of the covariance matrix of Θ ˆ=θ ˆ1,…,θ ˆGT. We can then construct a test statistic Q: (5) Q=nCθ ˆTCΣ ˆCT−1Cθ ˆ.

As n → ∞, the test statistic Q approaches a central chi-squared distribution with degrees of freedom υ under the null hypothesis in Eq. (4). Under the alternative hypothesis, Q approaches a noncentral chi-squared distribution with the same degrees of freedom υ and noncentrality parameter λ. For details of this test under the general SMART design, see Zhong et al. (2019). On the other hand, let θj and θk be the values of two strategies j and k, where j ≠ k. We can then conduct a pairwise test with the following hypotheses: (6) H0:θj=θkversusH1:θj≠θk,wherej,k=1,…Gandj≠k.

In analysis, the investigator can calculate the p-value of a pairwise test and compare it to a prespecified significance level α, adjusted for multiplicity. Both the global and pairwise tests control the overall significance level strictly, although the pairwise test is relatively conservative for large numbers of strategies. For example, eight strategies are embedded in the alcohol addiction trial (Fig. 2A). Suppose we choose to control the overall significance level at 0.05; a pairwise test then needs to have a p-value of less than 0.05∕28 = 0.0018 to claim overall statistically significance. Alternatively, we can construct a series of simultaneous confidence intervals: θ ˆj−θ ˆk±δjk×σ ˆjknjk,wherej,k=1,…Gandj≠k,

and then compare the intervals to zero. To control the overall confidence level at 1 − α, we need to adjust δjk as follows: (7) PZ≤δjk=1−αnp

where np=G−1×G2 is the number of contrasts. In the alcohol addiction trial, G = 8 and we thus obtain the value np=7×8÷2=28.

Similarly, the sample size calculation of a SMART is based on either a global test or a pairwise test, depending on the investigator’s preference and prior knowledge. In practice, an investigator can calculate the sample size of a SMART based on a global test according to the following steps.

Step 1: Determine the design diagram, set up the randomization probabilities, and calculate the degrees of freedom υ: (8) υ= ∑t=1TN At− ∑t=1TN Ot+1,

where NAt is the total number of treatment options and NOt is the total number of baseline/intermediate response categories at stage t in the design diagram.

Step 2: For a prespecified type I error rate α and a targeted power 1 − β, determine the noncentrality parameter λ∗ required under the alternative hypothesis of Eq. (4) by solving (9) χυ,1−β2λ∗=χ υ,α20,

where χυ,1−β2λ∗ denotes the (1 − β)th percentile of a noncentral chi-squared distribution with υ degrees of freedom and noncentrality parameter λ∗.

Step 3: For the given randomization probabilities, expected baseline/intermediate response rates, and sequence-specific means and variances of the primary outcome, calculate the targeted strategy values Θ∗ and its covariance matrix Σ∗ so that standardized overall effect size, Δ, can be calculated according to (10) Δ=CΘ∗TCΣ∗CT−CΘ∗.

Step 4: The total number of patients needed for a SMART is (11) n=λ∗Δ.

Alternatively, we can calculate the sample size based on a pairwise test according to the expected standardized difference between two ATS values of interest, as in a classic two-arm RCT. The parameter values needed to calculate the standardized difference can be obtained from Θ∗ and Σ∗ in Eq. (5). Note that a pair of strategies embedded in a SMART do not necessarily have to be independent, and the covariance varies by the design structure, randomization probabilities, intermediate response rates, and sequence-specific primary outcomes.

R functions

The newly developed SMARTAR package includes five primary functions: smartsize, getncp, seqmeans, atsmeans, and smartest (see Table 2). These functions provide important statistical results from the design to the analysis of a SMART for selecting the best ATS from a family of candidates. The first step of a SMART is to estimate the sample size required to achieve a targeted statistical power for a valid inference of comparison under the strict control of false positive findings. The smartsize function helps conduct sample size calculations in two different manners, using either a global test or a pairwise test. This function allows one to control design parameters, such as the targeted type I error rate, statistical power, the type of primary outcome, the method of estimation, and the type of test. An important parameter in powering a SMART with a global test is the noncentrality parameter λ∗ in (11), which can be obtained by solving Eq. (9). Generally, smaller type I error, greater statistical power, and larger degrees of freedom (reflecting the number of treatment options and the number of intermediate response categories) require a larger λ∗, and hence a larger sample size. The getncp function returns the value of the noncentrality parameter.

The SMARTAR package provides two functions, seqmeans and atsmeans, to summarize descriptive statistics at the treatment sequence level and at the ATS level. Both functions output summarized graphics for visualization. A SMART provides data consisting of multiple treatment sequences O ¯T,A ¯T,Y. The sequence a patient will follow depends on the results of randomization and baseline/intermediate evaluation (i.e., called the tailoring variables). Only one sequence can be observed for a patient who completes the trial. The seqmeans function returns the descriptive statistics of all the treatment sequences embedded in the SMART. For each sequence, it gives information about the definition of the treatment sequence (e.g., stage-1 treatment, intermediate response, and stage-2 treatment for the alcohol addiction trial example), the number of patients, the mean, variance, and standard deviation of the primary outcome. This function can output the design diagram and summary graph of the descriptive statistics (i.e., a boxplot for a continuous outcome and a bar chart for a binary outcome). Alternatively, the function atsmeans gives descriptive statistics corresponding to the ATSs, including those identified in a SMART dataset as defined by Eq. (3), estimated values θ ˆ with confidence intervals, and the estimated variance–covariance matrix Σ ˆ of these strategy values. This function also provides a forest plot to show all the estimated strategy values with confidence intervals.

Other than descriptive statistics, a valid conclusion of a SMART should be made based on the inference results of comparisons. Using the smartest function, the user can obtain the results of both global and pairwise comparisons, which are crucial for the final analysis report.

Illustration with a real data example

In this section, we illustrate how to apply the primary functions of SMARTAR using example data from a depression management trial. The illustration starts with the example followed by the data format applied to these functions. We then demonstrate the sample size calculation for the trial design. Afterward, we illustrate how to analyze the SMART data, including exploratory data analysis, descriptive statistics, hypothesis tests, and the use of simultaneous confidence intervals.

CODIACS trial

Cheung, Chakraborty & Davidson (2015) analyze SMART data for a subset of patients enrolled in a Comparison of Depression Interventions after Acute Coronary Syndrome (CODIACS) trial to determine further which stepped care depression management strategy should be used and which discontinued in an implementation stage. A specific task can thus be formulated as selecting the best adaptive depression management strategy for further practice, based on a reduction of Beck Depression Inventory (BDI) scores six months after enrollment, the primary endpoint of the trial. Higher BDI scores are known to be linked to more severe depression symptoms, and thus a strategy leading to a greater BDI reduction indicates a better treatment effect. In other words, the value of a depression management strategy, in this case, is the expected reduction of the BDI at six months. Each ATS, in this case, would adapt to an intermediate response, defined as a BDI reduction of at least three units during the eight-week step period. Two treatment options, medication (MED) and problem-solving therapy (PST), are considered at baseline and potentially at the end of the step period for modification. In this article, we use the CODIACS data as an example to demonstrate the usage of the primary functions in SMARTAR.

Data format

The SMARTAR package requires inputting wide-formatted data of a T-stage SMART with variables O1,A1,O2,A2,…,OT,AT,Y.

Each row of data represents the record of a patient who has completed the trial. If the stage-1 treatment is not adaptive to baseline information, O1 will be eliminated from data and the variables become A1,O2,A2,…,OT,AT,Y.

Below are examples of R code and the corresponding results in the R console. We first attached the package, retrieved the data named codiacs, and then printed the summary of variables and first several rows of this dataset. Here, A1∈0,1 and A2∈0,1 are the stage-1 and stage-2 treatment variables, respectively, with 0 denoting MED and 1 denoting PST. Considering that A1 is not adaptive here, there is no O1 in this dataset. The intermediate outcome O2∈0,1 is 0 for no response and 1 for response, and Y is the continuous primary outcome of a BDI reduction at six months. Although this dataset provides a patient indicator variable ID and treatment sequence indicator variable group, they are not necessary for this R package, because the functions will identify treatment sequence by the values of As and Os. If the variables names in the original data differ from the above, the user needs to rename them before running the functions.

Sample size calculation

Sample size calculation is key statistical work for controlling the budget, enrollment period, and study duration of a SMART. The smartsize function allows one to power a SMART trial flexibly. By setting the logical operator global=TRUE, we can calculate the sample size based on the global test with the hypothesis in Eq. (4). Otherwise, the operator outputs the results based on the pairwise test. A user can also set the targeted type I and II error rates via the two arguments alpha and beta, respectively, when calling this function. By default, alpha=0.05 and beta=0.20. The family argument in this function allows the features of the primary outcome to be specified for calculation, where family=“gaussian” for a continuous primary outcome and family=“binomial” for a binary outcome. The default is family=“gaussian”. Users can also control the calculation for two different methods, method=“Gest” for G-computation estimation (default) and method=“IPW” for the IPW estimation.

The R code and results of powering the CODIACS trial by the smartsize function are shown at the end of this section. The investigator is interested in a SMART with two stage-1 treatments, a binary intermediate outcome, and two stage-2 treatments at each decision point. The randomization probabilities are set to be 0.50 for each treatment option. To power the trial based on the global test (part 1), suppose the expected means and standard deviations of the primary outcome (i.e., BDI reduction at six months) for the eight treatment sequences are (1,8,11,10,9,6,20,12) and (10,10,10,10,10,10,10,10), respectively; an effect size Δ = 0.1213 can then be calculated by formula Eq. (7) and thus coded in the function as delta=0.1213. According to formula Eq. (8), the degrees of freedom are coded as df = 5. Therefore, the key information of the sample size calculation is the output in the R console. The noncentrality parameter of the chi-squared distribution under the alternative hypothesis in Eq. (4) is denoted NCP. In this case, we have NCP=12.8249. The effect size and degrees of freedom are delta and df, respectively, and N is the total sample size required.

Alternatively, an investigator can use the argument sim to input a sequence information matrix (SIM), which is a data frame containing all the required sequence-specific information for calculating the effect size and degrees of freedom. Part 2 of the example code shows the SIM and how to input it into smartsize. The data frame SIMatrix in this example includes the values of a series of variables {SEQ,A1,PI1,O2,P2,A2,PI2,MEAN,SD}, where SEQ is the treatment sequence indicator, A1 is the stage-1 treatment, PI1 is the stage-1 randomization probability, O2 is the intermediate response measured at the end of stage 1, P2 is the intermediate response rate, A2 is the stage-2 treatment, PI2 is the stage-2 randomization probability, MEAN is the sequence-specific mean, and SD is the sequence-specific standard deviation of the primary outcome. Note that Part 2 yields the same sample size calculation as Part 1, although the information is input into the function differently. In practice, the investigator first specifies the design structure of a SMART, based on the question of interest, clinical practice, and available treatment options. The investigator then makes the assumptions about the intermediate response rates, means, and standard deviation for each treatment sequence, based on prior knowledge. Using the SIM in Part 2 is appealing for clinical investigators who are not experts in statistical calculation. The results show that, for a SMART design with a design diagram as in Fig. 3A and applied equal randomization probabilities at each decision-making point, if the sequence-specific means and standard deviations of the primary outcome are expected to be the values in the SIM in Part 2, a total sample size of 106 patients helps achieve 80% power of rejecting the null hypothesis that all eight ATSs have the same values at a significance level of 0.05. A statistician who is familiar with SMART design can first calculate the effect size and then input it into the function via delta. The effect size is an essential measure of SMART design, since we can compare different possible designs by their calibrated effect sizes in the calibration.

Figure 3 (A) Design diagram and (B) box plots of the sequence-specific average BDI reduction of the CODIACS trial, as output from the function seqmeans.

In some cases, the investigator is particularly interested in comparing a pair of strategies. By specifying global=FALSE, smartsize can power a SMART based on a pairwise test, as shown in Part 3 of the example code. When powered by a pairwise test, the function inputs the standardized effect size of the pairwise test and returns the numbers of patients included in each ATS of the comparison, assuming equal sample sizes for the two strategies. For example, the results in Part 3 show that, if we would like to achieve 80% power in detecting a standardized difference between the two strategies of 0.90, we need to recruit 20 patients for each ATS in the planned SMART. This method is similar to powering a classic two-arm RCT based on t-tests. While the user needs to recalculate the total sample size based on the design structure. For more technical details on this method, see Murphy (2005).

	

Noncentrality parameter of chi-squared test statistic

An important element of inference related to global tests in SMART is the noncentrality parameter λ∗ of chi-squared test statistics. The function getncp helps obtain this value. In calling this function, the user needs to input the degrees of freedom, given the design structure of SMART, type I and II errors. Since no formula can calculate the value of λ∗ directly, this function starts searching for the value by calculating the left-hand side of Eq. (7), based on a prespecified value, and then approaches the targeted value until a stopping criterion is reached. In this case, the stopping criterion is the maximal absolute difference between the left- and right-hand sides in Eq. (9). In practice, the user can set the initial value of λ∗ by the numeric argument start and define the maximum absolute difference by d. Example 3 of the R code and the results shows the result of calling the function getncp in the CODIACS trial. Given the design structure, we calculate the degrees of freedom and input it by df = 5. We set the targeted type I and II errors as alpha=0.05 and beta=0.20, respectively. We then give an initial value of λ∗ with start=5 and the maximal acceptable difference d=0.0001. Therefore, the computer starts the search process at λ∗ = 5 and finally returns the value of the noncentrality parameter as 12.8249. Note that, when the getncp function is used, the start value of λ∗does not affect the results of the search, although a start value closer to the result leads to a shorter search process. On the other hand, the value of d affects the accuracy of the search. Generally, a smaller maximal acceptable difference leads to better accuracy.

List 3. R code and results: Example 3

>  getncp(df = 5,alpha=0.05,beta=0.20, d = 0.0001,start=5)

[1] 12.8249

Descriptive statistics

Once the SMART data are collected, it is natural to summarize the features of the data by treatment sequence. The seqmeans function can provide these results. By specifying the argument family, it adjusts the calculation for continuous (family=“gaussian”) and binary (family =“binomial”) primary outcomes. The argument digits controls the number of rounding decimal places for the final outputs. By default, this function returns calculation results without rounding. The argument plot allows the output graphs to be selected, including the design diagram (plot= “d”) and outcome summary graphics (plot=“s”), which are box plots for a continuous outcome and bar charts for a binary outcome. Example 4 of the R code and results shows the R program for calling the function seqmeans using the CODIACS data and the outputs. When calling this function, we input a data object named codiacs and specify the type of outcome (i.e., BDI reduction) to be continuous by setting family=“gaussian”. Meanwhile, we choose to output the design diagram with plot=“d”. The Appendix provides a detailed summary of all the arguments. In the output data frame, the SEQ column gives the indicators of all the treatment sequences embedded in this dataset. Each sequence is defined by the values A1,O2,A2 in the next three columns. The columns N, MEAN, VAR, and SD present the number of patients, the average value, variance, and standard deviation of the primary outcome corresponding to each treatment sequence. From these outputs, we know that the codiacs data include eight two-stage treatment sequences, defined as (A1,O2,A2), where A1=(0,1), O2=(0,1), and A2=(0,1). The output also gives the number of the patients and the mean, variance, and standard deviation of the primary outcome for each sequence. For example, the first row shows that 25 patients received medication at stage-1 (i.e., coded as 0), failed to respond (i.e., O2=0), and then continued to receive medication at stage-2 (i.e., A2=0). The average reduction in the BDI reduction at six months for these patients is 1.32, with a variance of 50.48 and a standard deviation of 7.10.

	

For the results of specifying plot=“d”, we obtain the design diagram in Fig. 3A. The seqmeans function also provides a series of arguments that allow for the customization of the output graphs, such as the title, label, legend, colors, and addition of a reference line. By specifying pch=c(19,15) and legend=c(“Intermediate evaluation”, “Stage-specific treatment”), the user can separate the treatment assignment points and intermediate evaluation points into different shapes (i.e., a dot for treatment and a diamond for intermediate evaluation). We also add the title of Fig. 3A by setting title=“(A) Design diagram of CODIACS trial” to inform the readers. Alternatively, specifying plot=”s” yields the box plot of the primary outcome of all eight treatment sequences in Fig. 3B. In this case, we specify color=“yellow” and title=“(B) Primary outcome by treatment sequence (A1,O2,A2)”. We also set the labels xlab=“Treatment sequence” and ylab= “BDI reduction”. A reference line indicating the average BDI reduction at six months across all trial participants is added to the figure with reference=TRUE. When the primary outcome is a binary variable, we can set family=“binomial” and the function replaces the box plots with bar charts.

The SMARTAR package also provides strategy-level descriptive statistics using the function atsmeans. By specifying an argument method=“Gest” or method=“IPW”, we can choose the method of estimation to generate the results. This function also provides options to control the calculations and outputs. For example, the logical operator common allows the pooled variance (common=TRUE) or sequence-specific variances (common=FALSE) to be used in the estimation. Another logical operator, conf, returns the confidence interval for each estimated ATS value. We can control the confidence levels by specifying the value of the argument alpha and output a forest plot of ATS values by setting the option plot=TRUE. By default, alpha=0.05 and the function automatically calculates the 95% confidence intervals.

Figure 4 R code and output of the astmeans function.

We show an example of R code calling the function atsmeans using the CODIACS data and the outputs in Example 5 of R code and results. We choose to calculate the estimated strategy values by the G-computation method (method=“Gest”) and output their 95% confidence intervals (conf=TRUE, alpha=0.05). We also set plot=TRUE to generate the forest plot. Therefore, the outputs start with brief explanations of the results at the top. The first output data frame gives the estimated strategy values ($value). Each row of this matrix corresponds to a specific depression management strategy. The first column of the output in List 5, ATS, is the strategy indicator. The columns d0, d00, and d01 are the stage-1 treatment and the stage-2 treatments for non-responders and responders under the strategy indicated by the ATS. The total number of patients who followed the strategy is denoted by N, and the value and se columns denote the estimated strategy value and its standard error. The lower and upper limits of the 95% confidence interval are also shown. The second output data frame is the estimated variance–covariance matrix of the strategy values. Figure 4 shows the estimated values of all eight depression management strategies with 95% confidence intervals. For example, from rows 2 and 3 of $value, we know that there were 30 patients in codiacs under the ATS with indicator 2, in which patients received medication at stage 1, continued to receive medication at stage 2 if responsive, or switched to PST otherwise. The average BDI reduction at six months under this ATS is estimated to be 3.33, with a standard error of 1.24. The 95% confidence interval of this estimated strategy value is (0.90, 5.76). Similarly, there were 26 patients under ATS 3, whose strategy value is estimated to be 10.69 with a standard error of 0.64. The 95% confidence interval is (9.44, 11.95). From the variance–covariance matrix output, we know that the covariance between the estimated values of ATSs 2 and 3 is 0.0065. Note that the variance–covariance matrix is a block diagonal matrix with the upper left block corresponding to the strategies {1,2,3,4} and the lower right block corresponding to strategies {5,6,7,8}. Graphically, the estimated values of two strategies that are at least partially overlapping will have non-zero covariance; otherwise, they will be independent of each other.

	

Statistical comparison

In this section we show an example of R code and its results calling the function smartest using the CODIACS data. The outputs first give a brief explanation of the results at the top and then provide three parts of the results, labeled $Strategy, $Global.test, and $Pairwise.comparisons. The first part (i.e., $Strategy) lists all the strategies embedded in the SMART, their definitions, and the number of patients who followed each strategy, which gives a snapshot of the candidates for comparison. The second part (i.e., $Global.test) provides the results of a global test under the hypothesis in Eq. (4), including the total sample size, the number of ATSs, degrees of freedom, the value of test statistic, and the p-value. In this case, a global test among the eight strategies using the CODIACS trial data of 108 patients yields a test statistic Q = 36.03. Under the null hypothesis of Eq. (4), Q follows a central chi-squared distribution with degrees of freedom ν = 5. We therefore obtain a p-value less than 0.05 and thus claim there is sufficient evidence to reject the null hypothesis that all eight depression management strategies lead to the same level of BDI score reduction at six months. This test can serve as a gate-keeping test for ATS selection in the SMART. In this case, selection only occurs when there is an adequate level of statistical evidence concerning the difference among different treatment strategies, whereas, if the result fails to reject the null hypothesis in Eq. (4), a conclusion of no overall statistical difference would be made and no further analysis would be conducted.

The third part (i.e., $Pairwise.comparisons) provides the results of pairwise comparisons. Each row in this part corresponds to a pair of strategies, indicated by the label in the first column. The second, third, and fourth columns are the point estimates of the difference between the two strategy values and its 1−α% confidence limits. The fifth and sixth columns show the test statistics and p-values, respectively. The user can control the outputs with a series of arguments. This function also contains the option ntest, which allows the number of pairs of comparison np for multiplicity adjustment to be set. The user can also control the confidence level by setting the value of alpha. These options provide flexibility for analysts to answer different research questions. In this case, we set adjust=“Bon”, which returns the simultaneous confidence intervals adjusted for Bonferroni correction; otherwise, it gives the confidence intervals without multiplicity adjustment. For example, the second row shows the results of comparing strategies 1 and 3. The difference between the two estimated values is −4.43, with a 95% simultaneous confidence interval of (−7.39, −1.46). This confidence interval does not cover zero, and thus we claim an overall difference from the comparison between these two strategies. A pairwise test of this pair returns a test statistic Z =  − 4.66 and a p-value of 0.00. Although compared to the global test, pairwise tests are more conservative due to the multiplicity adjustment, they do provide the information on the value differences between any pair of strategies and are thus important in analyzing SMARTs.

	

Discussion

Statistical research in personalized medicine has attracted increasing interest over the past 20 years. Although the concept of personalized medicine varies by the literature and research communities, the core idea involves individual-level information as the fundamentals of clinical decisions (Kosorok & Moodie, 2015). An ATS provides opportunities to adjust the treatment selection sequentially, according to the clinical history and individual responses of each patient to previous treatment, thus fits the paradigm of personalized medicine. Although more and more statistical researchers are contributing to this field, there is still a gap between theoretical statistical research and clinical trial practice. A key problem is that there is no user-friendly statistical package available for those non-statistical methodology researchers to implement these methods in practice. Therefore, we developed the SMARTAR package. This package was designed specifically for clinical investigators and data analysts to conduct statistical work in the pipeline of a clinical trial with a SMART design.

Traditionally, researchers calculate the sample size of a SMART using a pairwise comparison approach, due to the relatively simple calculation (Murphy, 2005; Dawson & Lavori, 2010). An investigator therefore needs to first select a pair of ATSs and then calculate the standardized difference between them in an ad hoc manner. Statisticians thus need to first derive the formulas of two strategy values and the variance of the value difference, calculate the sample size required for the two targeted strategies, and then obtain the total number of patients needed to enroll in the trial. This method reamins rather conservative when the design structure of a SMART becomes more complex, because the multiplicity adjustment will reduce the targeted type I error. Alternatively, researchers proposed powering a SMART based on an omnibus test (Ogbagaber, Karp & Wahed, 2016). Zhong et al. (2019) derived the sample size calculation based on an omnibus test under a general SMART design. This method can be applied to SMARTs with various design structures, and simulations show that the gate-keeping approach based on this method has advantageous properties (i.e., strict control of type I error and high accuracy of the selection of the true best ATS) in selecting the best ATS. The primary functions of SMARTAR support all these methods.

Because of the increasing interest in this field, newly developed statistical methods are published in this field every year. Cheung, Chakraborty & Davidson (2015) analyze CODIACS data with the goal of selecting the best two-stage ATSs using an iterative regression-based Q-learning method. The optimal depression management strategy therefore would start with PST and switch to medication, regardless of the intermediate outcome, which is consistent with our finding using the gate-keeping approach by implementing the smartest function. In practice, the gate-keeping approach is preferred because it provides a p-value that helps directly control false positive findings. Due to the limited resources and time, we focus on those methods commonly used in trial practice, but not those in observational studies, such as Q-learning (Murphy, 2006) or A-learning (Zhao et al., 2015). Considering the targeted users of this R package, we believed it is the optimal solution in software development.

Future updates of this package will keep focusing on clinical trial design and analysis purposes. We plan to update it by adding newly developed statistical methods for trial practice every two years. The updated version will be posted and available from CRAN and GitHub.

Conclusion

SMARTAR is a comprehensive statistical software package for designing and analyzing clinical trials with a SMART design. We introduce the functions of this R package and illustrate how to apply these functions using examples with real data. This package will help clinical investigators and statisticians in practice.

Computational details

The results in this paper were obtained using R 3.6.1 with the SMARTAR 1.1.0 package. R itself and all packages used are available from the Comprehensive R Archive Network (CRAN) at https://CRAN.R-project.org.

Supplemental Information

Supplemental Information 1 R source package

Click here for additional data file.

Supplemental Information 2 Example R code used to replicate all the examples

Click here for additional data file.

We would like to thank Ms. Alica Brewer for providing editial help in this article. Drs. Pranav Pandit, Zhian Kamvar, and William Love helped review the R package.

Appendix 1: Summary Document for Arguments

This appendix provides the detailed explanations and the options of each argument for the five key functions of the SMARTAR package, namely, smartsize, getncp, seqmeans, atsmeans, and smartest.

Function 1: smartsize

This function provides the result of sample size calculation for a sequential multiple assignment randomized trial (SMART). The explanations and the options of the arguments included in this function are as follows.

• alpha: numeric. Specify the targeted type I error of a sample size calculation. The default is alpha=0.05.

• beta: numeric. Specify the targeted type II error of a sample size calculation. The default is beta=0.20, which leads to a targeted statistical power of 0.80.

• global: logic. Specify the test of a sample size calculation. If global=TRUE, the sample size calculation will be based on the global test; if global=FALSE, the sample size calculation will be based on a pairwise test. The default is global=TRUE.

• family: specify the type of primary outcome to be analyzed. This argument includes two options, “gaussian” for a continuous primary outcome and “binomial” for a binary outcome. The default is family=“gaussian”.

• df: integer to specify the degrees of freedom for the chi-squared test statistics of the global test.

• delta: a positive real number to specify the effect size for sample size calculation. The user can choose to input the effect size of the global test, as defined by Zhong et al. (2019), in this argument, given global=TRUE, or input the standardized difference of the two strategy values, given global=FALSE.

• sim: sequence information matrix for calculating the effect size of the global test. If users choose to power the trial by the global test, they can just input the sequence information matrix directly, instead of calculating the effect size themselves. This matrix includes all the sequence-specific information. Specifically, the matrix includes N rows, where N is the total number of sequences embedded in the SMART. Each row contains information for the sequence indicator (SEQ), tailoring variables (Ot), treatment variables (At), intermediate response rate (PT), randomization probability (PIt), sequence-specific mean (MEAN), and standard deviation (SD) of the primary outcome.

Function 2: getncp

This function returns the value of the noncentrality parameter of the chi-squared distribution for the test statistics of the global test under the alternative hypothesis. The theoretical value of the noncentrality parameter is a function of type I error, type II error, and degrees of freedom. The explanations and the options of the arguments included in this function are as follows.

• df: integer to specify the degrees of freedom for the chi-squared test statistics of the global test.

• alpha: numeric. Specify the targeted type I error of the sample size calculation. The default is alpha=0.05.

• beta: numeric. Specify the targeted type II error of the sample size calculation. The default is beta=0.20, which leads to a targeted power of 0.80.

• d: critical value of the distance of the search procedure. The search for the non-centrality parameter value stops at the absolute distance between the actual power and the target power less than the value of d. The default value of d = 0.0001.

Function 3: seqmeans

This function outputs all the treatment sequences embedded in a SMART dataset, summarizes all the sequence-specific descriptive statistics and graphs, and provides a design diagram of the SMART. Each treatment sequence is indicated by the values of the stage-specific tailoring variables and the treatment variables corresponding to a patient who follows this sequence. The explanations and the options of the arguments included in this function are as follows.

• data: input data frame of the sequential multiple assignment randomized trial (SMART) data to be analyzed.

• family: type of primary outcome to be analyzed. This argument includes two options, “gaussian” for a continuous primary outcome and “binomial” for a binary outcome. The default is family=“gaussian”.

• plot: type of output figure. Here, set plot=d for design diagram of the SMART and plot=s for the figure to summarize the descriptive statistics by the treatment sequence. Specifically, plot=s will output box plots when the primary outcome is continuous (i.e., family=“gaussian”) and bar plots when the output is binary (family=“binomial”). The default is plot=d.

• digits: integer indicating the number of decimal places for the descriptive statistics in the output, including the sequence-specific mean, variance, and standard deviation. By default, digits=NULL and the function returns descriptive statistics without rounding.

• color: characters indicating the color of the boxplots and barplots. The second default is c(“yellow”,“forestgreen”). For a continuous outcome (i.e., family=“gaussian”), this option allows the color of the boxplots to be selected, and the default is color=“yellow”. For a binary outcome (i.e., family=“binomial”), this option controls the colors of the bar plots, with one for the outcome of “yes” and one for “no”. In this case, the user needs to input two characteristics. By default, color=c(“yellow”, “forestgreen”).

• pch: two integers indicating the point shape of the tailoring variables and the treatment variables in the design diagram. It allows the points indicating baseline/intermediate evaluations and treatment assignments in the design diagram to be separated. The first integer specifies the shape of the treatment assignment points; the second integer specifies the shape of the evaluation points. The default values are pch=c(19,15), indicating a square for evaluations and dots for treatment assignments.

• title: characters to specify the title of the output figure.

• ylab: characters to specify the label of the vertical axis of the output figure.

• xlab: characters to specify the label of the horizontal axis of the output figure.

• legend: characters to specify the legend of the design diagram of the SMART. The default is legend=c(“Evaluation”,“Treatment”).

• reference: logic argument to add a reference line to the graph of descriptive statistics of the primary outcome. The value of the reference line is equal to the average of all sequence-specific means. If TRUE, add a reference line (mean of outcome) to the box plot of the primary outcome; otherwise, do not add a reference line. The default is reference=FALSE.

Function 4: atsmeans

This function outputs all the ATSs embedded in a SMART dataset, with the estimated strategy values and their estimated asymptotic variance–covariance matrix. The explanations and the options of the arguments included in this function are as follows.

• data: input data frame of the SMART data to be analyzed.

• family: type of primary outcome to be analyzed. This argument includes two options, “gaussian” for a continuous primary outcome and “binomial” for a binary outcome. The default is family=“gaussian”.

• method: specify the method to calculate the estimated strategy values. The users can select one of two common methods, with “Gest” for the G-computation method and “IPW” for the Inversed Probability Weighting method. By default, method=“Gest”.

• digits: integer indicating the number of decimal places for descriptive statistics in the output, including the sequence-specific mean, variance, and standard deviation in the outputs. By default, digits=NULL and the function returns descriptive statistics without rounding.

• common: logic. This argument indicates whether the variance–covariance matrix of strategy values will be calculated based on the pooled variance across all the treatment sequences or not, with TRUE for pooled variance and FALSE for sequence-specific variances. The default is common=TRUE.

• conf: logic, whether the confidence intervals of the estimated strategy values are output or not. If conf=TRUE, the estimated strategy values with confidence intervals will be output. The default is conf=TRUE.

• alpha: numeric. Specify the confidence level of the estimated strategy values at (1 - α)%, given conf=TRUE. The default alpha = 0.05.

• plot: logic. If plot=TRUE, the function will output the boxplots of all the estimated strategy values with (1 - α)% confidence intervals. The default is plot=TRUE.

• title: characters to specify the title of the output graphs. The default is “Strategy values with confidence intervals”.

• color: characters indicating the color of the graph. The default is color=“forestgreen”.

• ylab: characters to specify the label of the vertical axis of the output figure. The default is “Strategy value”.

• xlab: characters to specify the label of the horizontal axis of the output figure.

• xtext: characters to specify the text of the horizontal axis of the figure.

• pch: integer to specify the shape of the points in the graphs. The default is pch=15.

• cex: integer to specify the amount by which plotting symbols should be magnified. The default is cex=2.

• lwd: integer indicating the line width. The lines refer to the width of the confidence interval. The default is lwd=1.

• ylim: integers to specify the maximum and minimum values of the y-axis.

• mar: numerical vector of the form c(bottom, left, top, right), which gives the number of lines for the margin to be specified on the four sides of the plot.

• cex.axis: magnification to be used for axis annotation relative to the current setting of cex.

• line: specification of a value for a line that overrides the default placement of the label of the horizontal axis of the graphs.

Function 5: smartest

This function provides the results of statistical tests comparing ATSs based on both global and pairwise tests. The explanations and the options of the arguments included in this function are as follows.

• data: input data frame of the sequential multiple assignment randomized trial (SMART) data to be analyzed.

• family: type of primary outcome to be analyzed. This argument includes two options, “gaussian” for a continuous primary outcome and “binomial” for a binary outcome. The default is family=“gaussian”.

• method: method to calculate the estimated strategy values. The users can select one of two common methods: “Gest” for the G-computation method and “IPW” for the Inverse Probability Weighting method. By default, method=“Gest”.

• digits: integer indicating the number of decimal places for the descriptive statistics in the output, including the sequence-specific mean, variance, and standard deviation. By default, digits=NULL and the function returns descriptive statistics without rounding.

• common: logic. This argument indicates whether the variance–covariance matrix of strategy values will be calculated based on the pooled variance across all the treatment sequences or not, with TRUE for pooled variance and FALSE for sequence-specific variances. The default is common=TRUE.

• alpha: numeric. Specifies the confidence level of the difference between each pair of estimated strategy values at (1 - α)%. The default alpha=0.05.

• adjust: argument allows calculation of the confidence intervals of the strategy value difference corrected for the multiplicity adjustment. By specifying adjust = “Bon”, the confidence interval is adjusted for Bonferroni correction. The default is adjust = NULL, which means that the estimated strategy value differences are not adjusted for multiplicity comparisons.

• ntest: integer to specify the number of pairwise comparisons adjusted for Bonferroni correction, given adjust = “Bon”.

Appendix 2: Simulation Study

Having introduced the background, contents, and usage of the SMARTAR package in the manuscript, we compared the results generated by the functions belonging to this R package using simulation. The simulated results were compared with the corresponding theoretical results and the simulation results reported by Zhong et al. (2019) that were not based on the SMARTAR package.

SMART designs

Figures 5A and 5B give two design structures of two-stage SMARTs considered in the simulation. Figure 5A is design structure 1 (DS1), under which patients were first randomized to two treatment options at stage 1. At the end of stage 1, patients were classified as responders or non-responders. Patients were then randomly assigned to two treatment options in stage 2, according to different histories of stage-1 treatments and intermediate outcomes. As results, there are 8 ATSs embedded in DS1. Figure 5B shows the DS2. There were two treatment options in stage 1 and a binary intermediate response. However, responders continued receiving the treatment in stage 1, and only non-responders were randomized into two treatment options at stage 2. DS2 led to 4 ATSs. We considered balanced randomization (BR), under which the randomization probability for each treatment option was set to be 0.50, whenever there is an option of randomization. We fixed the sample size for each simulated data at N = 200, and the simulation was conducted based on 5,000 replicates.

Figure 5 Design structures considered in the simulation.

Outcome scenarios

We set the intermediate response rate for each simulated SMART data to be 1/3 for both A1 = 0 and A1 = 0. Given a subject’s treatment history and intermediate outcome (A1, O2, A2), the sequence-specific primary outcome was randomly generated based on a normal distribution with mean ϕ(A1, O2, A2) and variance σ2 = 100, where ϕ(A1, O2, A2) was specified by ϕA1,O2,A2=β0+β1A1+β2O2+β3A2+β4A1O2+β5A1A2+β6O2A2+β7A1O2A2

For A1, O2, A2 ∈ {0, 1}. The parameter β0,β1,β2,β3,β4,β5,β6,β7T

Figure 6 Value patterns of adaptive treatment strategies in the simulation.

was chosen so that the true values θ′s would follow the patterns displayed in Fig. 6. Under Value Pattern 1 (VP1), strategies with the same stage-1 treatment had the same values. Under VP2, the strategy values were uniformly higher if the stage-1 treatment was A1 = 1. Under VP3, the best strategy had stage-1 treatment A1 = 1 while the second best had stage-1 treatment A2 = 0, and so on and so forth, following an alternating pattern. The value of β′s were chosen so that the effect size of the global test was Δ = 0.05 and 0.10. For example, under VP1, setting β0 = β1 = … = β7 = 0 and β1 = 4.48 yielded the effect size Δ = 0.05. Details values of β′s for each value pattern are given in Table 3. In summary, two design structures (i.e., DS1, DS2), 3 value patterns (i.e., VP1, VP2, and VP3), and two effect sizes (i.e., Δ = 0.05 and 0.10) yielded a total of 12 scenarios in simulation.

Table 3 Two-stage adaptive treatment strategies defined in an alcohol addiction trial.

Design structure	Value pattern	Δ = 0.05	Δ = 0.10	
DS1	VP1	(0,4.48,0,0,0,0,0,0)	(0,6.33,0,0,0,0,0,0)	
DS1	VP2	(0,3.63,0,2.62,0,0,0,0)	(0,5.13,0,3.70,0,0,0,0)	
DS1	VP3	(0,1.86,0,3.73,−9.32,1.86,−0.93,0)	(0,2.64,0,5.82,−13.20,2.64,−1.32,0)	
DS2	VP1	(0,4.48,0,0,0,0,0,0)	(0,6.33,0,0,0,0,0,0)	
DS2	VP2	(0,0,0,2.88,12,0,0,0)	(0,0,0,4.13,17.70,0,0,0)	
DS2	VP3	(0,−1.21,0,4.82,4.82,1.21,0,0)	(0,−1.72,0,6.87,6.87,1.72,0,0)	
Notes.

* Details of design structures (DS) and value patterns (VP) are given in Appendix 2.

Results

Strategy value

We observed the estimated ATS values from the atsmeans function’s outputs when the sample size of simulated SMART was fixed at N = 200. Figure 7 shows the true value and the first 200 estimated values of each ATS in the 5000-replicate simulation under VP2 of DS2, when the global test’s effect size was Δ = 0.05 and 0.10. The bold solid line represents the true strategy values, which were (0.00, 1.92, 4.00, 5.92) for Δ = 0.10 and (0.00, 2.77, 5.90, 8.67) for Δ = 0.10. We can see that the estimated strategy values, calculated by the atsmeans function, always fluctuated around the corresponding true values, which indicated that the estimated strategy values outputted from the atsmeans function are unbiased estimates. The simulation results under other DSs and VPs led to the same conclusion. For details of other simulations, see GitHub (https://github.com/tonizhong/SMARTAR/tree/master/Simulation)

Figure 7 True and the first 200 estimated adaptive treatment strategy values in the simulation.

Table 4 Empirical statistical powers calculated based on the SMARTAR package in simulation and reported by Zhong et al. (2019).

Results obtained from	Design structure	Effect size Δ = 0.05	Effect size Δ = 0.10	
		VP1	VP2	VP3	VP1	VP2	VP3	
SMART package	DS1	0.6620	0.6674	0.6602	0.9444	0.9470	0.9474	
Zhong et al. (2019)	DS1	0.6700	0.6701	0.6720	0.9510	0.9330	0.9450	
SMART package	DS2	0.7550	0.7526	0.7698	0.9734	0.9762	0.9758	
Zhong et al. (2019)	DS2	0.7610	0.7570	0.7570	0.9770	0.9740	0.9760	

Actual type I error

We studied the actual type I error rates of the global test in simulation, defined as the proportion of rejecting the null hypothesis of Eq. (4) across 5,000 simulation replicates when the effect size Δ = 0, under both DS1 and DS2, using our R package’s smartest function. The significance level was set to be 0.05. The outcome scenario was generated by setting β = (0, 0, 0, 0, 0, 0, 0, 0)T, which led to the same value of 0 for all the ATSs. The simulations achieved type I error rates of 0.0478 and 0.0502 across 5,000 replicates under DS1 and DS2, respectively, close to the significance level of 0.05. These results were similar to those (i.e., 0.0480 for DS1 and 0.0500 for DS2) based on the simulations not relied on the SMARTAR package in Zhong et al. (2019).

Statistical power

We also verify the empirical statistical power of the global test, defined as the proportion of rejecting the null hypothesis of Eq. (4) across 5,000 simulation replicates when the effect size of a simulated data was Δ > 0, based on the smartest function. The first two rows of Table 4 provide the empirical powers obtained based on the smartest function in our simulations, and those not relied on this package in Zhong et al. (2019). We can see that the power calculated by the SMARTAR package was consistently close to the corresponding power reported in the previous study. Similar results were found across the 3 VPs (VP1, VP2, and VP3) and two effect sizes (0.05 and 0.10) when the design structure was DS2.

Additional Information and Declarations

Competing Interests

Author Contributions

Data Availability

The authors declare there are no competing interests.

Xiaobo Zhong conceived and designed the experiments, performed the experiments, analyzed the data, prepared figures and/or tables, authored or reviewed drafts of the paper, and approved the final draft.

Bin Cheng and Ying Kuen Cheung conceived and designed the experiments, authored or reviewed drafts of the paper, and approved the final draft.

Xinru Wang performed the experiments, analyzed the data, prepared figures and/or tables, and approved the final draft.

The following information was supplied regarding data availability:

Both the R package and the data used for the illustration example are available in the Supplementary Files.

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
