# Peer review of "SMARTAR: an R package for designing and analyzing Sequential Multiple Assignment Randomized Trials"

_PeerJ, doi:10.7717/peerj.10559_

## Round 0.1 · original submission · Major Revisions

For further consideration please identify your response to all the reviewer comments by line number, including the annotated comments provided in PDF by reviewer 1. Proofreading of the manuscript for English language is also required. Addressing both of these issues is required before acceptance.

·

Basic reporting

Since this is an article that presents a new R package, I have general comments about the structuring of the manuscript. Hence the review is below in the general comments section.

Experimental design

Since this is an article that represents a new R package, I have general comments about the structuring of the manuscript. Hence the review is below in the general comments section.

Validity of the findings

Since this is an article that represents a new R package, I have general comments about the structuring of the manuscript. Hence the review is below in the general comments section.

Additional comments

Authors present here an R package that they have developed for Sequential Multiple Assignment Randomized Trials (SMART). The overall manuscript is well developed, easy to read and the data is nicely represented. During this review, I went through the rOpenSci’s guidelines for reviewing a package too and checked if the manuscript fits in the review template provided by the rOpenSci community.
Here are my detailed comments. Within the introduction, the authors explain in the SMART trial approach. They illustrate how the approach is essential in developing trails for adaptive treatments with a couple of simple figures. Following that authors describe the current milieu of R packages that can be used for analyses related to SMART and briefly describe how the new package is advantageous over the others.
The authors describe in detail all functions presented in the package and that are required for the analytical pipeline of SMART. Besides tiny missing details, authors have described the functioning of the package very well. I would say that one important aspect of the case study is missing though that is the interpretation of results and outputs. To address those, I would recommend authors to first start with the Sample Size function (as the first step of SMART planning) followed by data visualization and statistical comparison. At each step, authors can explain the functioning of the package with code and data. This should then include discussion regards to the results.
For example: in lines, 307-311 authors say “In this case, conducting a global test among the 8 strategies 308 using the CODIACS trial data of 108 patients, it returned a test statistics Q = 36.03. Under 309 the null hypothesis, Q follows a centralized chi-squared distribution with degrees of freedom 310 ν = 5. As a result, we obtained a P-value less than 0.05 and claimed over statistical significant 311 difference among the eight depression management strategies in the CODIACS trial.” This should be followed by discussing the significance in terms of ATSs and their outcomes. I would recommend seeing Elith et al. 2008 A working guide to boosted regression trees. To follow the idea of explaining the biological interpretations of a package developed.
Another comment I would include is to develop extensive documentation. Including various options for arguments. Authors provide some explanations in the text, but centralized documentation is also essential as eventually users and developers would mostly refer to the package documentation. Similarly, help(SMARTAR) did not yield any documentation within my IDE while testing the package. There are additional small comments that I have noted while reviewing the package and they are attached below with the review.



Please see attached pdf document with a detailed review of the package based on rOpenSci’s guidelines.

·

Basic reporting

The basic reporting is detailed enough and adequate to follow along easily.

Experimental design

## Package Design

This package is on par with the design of its contemporaries, but it could use
some improvement that will improve the quality and make it stand out from other
packages. I will highlight a few things that struck me:

### Output

The output of the summary functions rounds the statistics before returning to
the user. Despite the fact that the "digits" option is available, I would
suggest to avoid this because it can be frustrating for users who do not want
their data rounded in the first place.

The output of smartest presents the $Strategy as a matrix instead of a data
frame. It's a small point, but it is more consistent to present this as a data
frame since each column represents variables of different modes.

### Tests

There are no formal tests in this package. As I mentioned before, this is on-par
with its contemporaries, but tests make sure a package provides both
reproducible and accurate results. Moreover, tests allow authors to refactor
code confidently without fear that it will break. I would recommend including a
test suite such as testthat or RUnit and writing tests that test known values
from published data sets and simplified scenarios. A very good resource on
writing tests is <https://r-pkgs.org/tests.html>

I would also strongly suggest the authors keep the package under version control
in a public repository (such as GitLab or GitHub) so that they can access
continuous integration tools.

### Documentation

The documentation provided in the package is enough to understand what it does,
but could use improvement. One suggestion is for the authors to write a vignette
describing a typical workflow. These are documents that live with a package that
are meant to describe how all the moving parts of the package work together.
A resource on those can be found here: <https://r-pkgs.org/vignettes.html>

The examples in the package should use the CODIACS data set or another real
data set. Users should not have to run ~30 lines of code before they run the
example function. Having examples of different scenarios would also allow the
authors to demonstrate the flexibility of the package.

### ChangeLog

One frustration I had with the package is that the version on CRAN was different
from the version provided here and there was no record of changes that I could
easily inspect. One suggestion I have is to include a NEWS or ChangeLog file in
the package that describes changes that the user should be aware of that include
different plotting options, new parameters, outputs, or documentation.

### Code style

I would recommend using the package "goodpractice" on your code to highlight
potential issues. I will highlight some important ones here:

In several of the functions there are conditional statements
(if statments) that use `&` instead of `&&`. All conditional statments should
use `&&` (note: this does not include `which()`). The reason for this is that
`&` will return a vector of any length, but `&&` will always return a vector of
length 1.

The authors should use `seq(ns)` instead of `1:ns` in their code.
The reason for this is because if ns is 0, then `1:ns` will be the vector
c(1, 0) instead of just 0.

The authors should use `TRUE` and `FALSE` instead of `T` and `F` as the latter
can be overwritten with user variables and may cause problems.

Validity of the findings

I cannot comment on the validity of the statistical methods used as I do not
have expertise on clinical design. The data and code provided replicate the results observed.

Additional comments

SMARTAR is a new R package that supports SMART analysis. While there are several
packages available for SMART analysis, SMARTAR differs in the following aspects:

- Supports Exploratory Data Analysis (L87)
- Provides graphical tools at the treatment sequence and strategy level
- Can generate SMART design diagram
- Can construct (Bonferroni-corrected) simultaneous confidence intervals for
ATS comparisons.
- Provides sample size calculation for both pairwise and global comparisons.

The authors identify that there are five other packages that perform some facet
of SMART analysis, but stress that SMARTAR provides functionalities beyond that
of the previous packages. The authors then take care to describe the input data
in detail and provide a use-case based on a real data set.

The package itself works as presented when installed via the supplementary
materials, but does not work from the version on CRAN, despite identical version
numbers. There are things that could be improved that would make this package
stand out from its contemporaries in the field of SMART design, including
testing, documentation, and de-duplication.


### Line changes

- L21--22: strike out "by far"
- L70: "DTReg" -> "DTRreg"
- L79: "limits" -> "limitations"
- L85: "is appealing to" -> "appeals to"
- L89--90: The sentence that starts with "It is by far..." is not needed.

·

Basic reporting

General notes:

The authors have created a useful R library that appears to address a need in randomized clinical trial design. The manuscript has been divided into the typical parts (Introduction, Materials & Methods, etc) though this is difficult to do efficiently with articles like this where the "study" being described is a piece of developed software. Additionally, the language is frequently awkward with incorrect and words and improperly used phrases. Some specifics are listed below, but there are too many examples in the entire paper to enumerate.

The "Real data example" appears around line 132, but there appears to still be more general functional explanation (lines 148, 171, 286, and equations 4-6). Especially with only 4 or 5 exported functions in the library, I believe it would be more clear to explain the library functions first, then provide examples of them in use later. This will allow clear sections laying out functions prior to involving real data.

Since the results of the CODIACS study is not the focus of the paper, it may be more appropriate to either explain how to use the results of the code in the discussion section. It would also be useful to compare how the package could have improved sample size selection, and how the the results produced by the the SMARTAR package compare to the results in Cheung et al. (2015)

The term "trialists" is an odd choice and in my experience not frequently used. In the discussion, the authors use the phrase "clinical investigators and data analysts" which is much more common parlance. Consider changing "trialist" to "investigators" or even simply "users" through out the document.

Other notes:

Introduction, first paragraph: Consider breaking this paragraph into several. The first paragraph should introduce the method, its use(s), its advantages, and perhaps begin the discussion about the analytic challenges of it presents. In the second paragraph, discuss more details and refer to the example. It seems odd to figure references in the first paragraph.

Line 61: "More details of ATS and SMART see [citation]" I believe this is should be "*For* more details see ..." This appears in several places, and it may be more appropriate to simply cite the respective articles in the preceding sentence.

Line(s) 109: The set notation here is somewhat confusing. For example A_1i = { CBT, MED } , but the i subscript indicates which treatment the ith individual received, not the set of possible . It appears this should be A_1 = { CBT, MED }, or A_1i is a subset of { CBT, MED }, or A_1i = CBT.

Line 149: "We first lunch the SMARTAR, retrieve the example data..." This should be "launch" instead of "lunch." A better term is "attach" the library, but the code is so basic that it doesn't really bear mention in the manuscript; same with following line.

Line 188: "We input a SMART data..." The word 'data' plural; this should read "we input SMART data ..." or "We input a SMART data object ..."

Experimental design

Based on the text and limited review of the included code, the design of the SMARTAR package appears to be sound. The example code runs in R 3.6.2 and returns results consistent with the manuscript text.

The 'seqmeans' function reports mean and variance to [digits] places. I would recommend reporting mean and std. deviation to the same number for significant figures, but providing more decimal places for the variance.

Validity of the findings

Again, the results reported by package appear to be consistent with the manuscript inputs and supplemental materials. However, it is somewhat difficult to assess the validity of the results without access to the 'unbuilt' code (the body of the library's functions are accessible via the body function, but it's much easier to read raw code). I would suggest the authors create a public-facing GitHub page to allow review and modification of the code for transparency. Also, an open-source license should be declared.

In the introduction, the authors mention several other packages designed to assist investigators in designing and analyzing SMART trials. The authors should compare and contrast the relevant output of the of their package to the results of the established packages.

Additional comments

Overall, the package developed by the authors appears useful and is consistent in quality with CRAN standards, but the language issues make manuscript very challenging to review. Additionally, given there are software packages that already exist to assist with these calculations, the performance of the current package should be compared to the results of applying the same data to other packages.

I would suggest having the paper reviewed and re-written with the assistance of a native English speaker to resolve these issues prior to the next round of review.

Other coding notes:
In the 'astmeans', the family and method arguments should be limited to a set of possible values. For example, the family argument could be defined using: astmean <- function(..., family = c("gaussian", "binomial") ). Arguments declared in this way will default to the first value in the concatenated set.

The authors may want to consider using informal S3 classes for their output. This allows the use of pre-existing generics like 'summary', 'print', and 'plot', while still allowing the constituent elements of the returned list objects. See Hadley Wickham's advanced R page, specifically http://adv-r.had.co.nz/S3.html

---

## Round 0.2 · Minor Revisions

Its my opinion that while the manuscript clarity has improved, there are comments and suggestions by reviewer 2 that will add to the manuscript's package usefulness to the user. Please address these comments and I look forward to moving this along.

·

Basic reporting

Authors have meticulously worked on the comments, improving the structure of function outputs, and improving version control. Vignettes provided online will be useful for package users improves the usability of the package. I like the way authors have explained functions with additional interpretation. The overall flow of the manuscript is also improved.

Experimental design

no comment

Validity of the findings

no comment

Additional comments

The authors have satisfactorily addressed all my comments and have improved the reproducibility of the package.

·

Basic reporting

The reporting is detailed and easy to follow.

Line 123: it is not necessary to state that the article ends with a discussion and conclusion.
Line 625: the "Strategy" output in list 6 still needs to be updated to the current output format.

Experimental design

I would like to thank the authors for taking the effort to add NEWS and implement tests for their package.

I believe the tests could stand for a bit more improvement. At the moment, the 64 tests take 400 seconds to run on my macOS and upon further inspection, none of the tests actually check that the output is correct, but instead rely on testing that there are no warnings generated by the code, of which there are three places that could generate such a warning.

I would strongly suggest to the authors to test that the outputs are equal to known quantities or at least within expected ranges.

Validity of the findings

No comment

Additional comments

No comment

---

## Round 0.3 · accepted · Accept

Thank you for addressing the reviewer comments. I did notice that one of the comments remaining had to do with validation of the package itself. Since this is not affecting the manuscript, rather is related to the package itself and its long term functioning within the R environment, and since I am satisfied with the necessary simulations to validate your analysis results, I am able to make an acceptance decision and leave it up to your team to pursue any validation within R. Congratulations and I look forward to using your package and recommending it to colleagues.

·

Basic reporting

no comment

Experimental design

I believe the authors had misunderstood my comments from the previous review. When I looked at the testing code for the package, no tests to check validity of output were added to the test suite. The added simulation study is a nice exercise, but it is not what was requested and not a substitute for adding tests for validity to the suite.

The simulation study is performed only once and tests the software as it is at a single piont in time, while the test suite is run every few days on CRAN and will make sure the package continues to produce correct results as the source code and the software it depends on changes.

I would point the authors to read Section 12.3 in "R Packages" by Wickham and Bryan for more information about the strategy for writing tests: https://r-pkgs.org/tests.html#test-tests and request once again that they put validation tests in their test suite.

Validity of the findings

no comment

Additional comments

The effort you have put in for testing and design of this package outpaces the efforts of the competing packages and I think you should be proud of that. Adding the verification tests to the suite will make your package even better and make it easier for you to refactor or update your code in the future.